# Dynamic Surface Topography for Thoracic and Lumbar Pain Patients—Applicability and First Results

**DOI:** 10.3390/bioengineering12030289

**Published:** 2025-03-13

**Authors:** Johanna Kniepert, Henriette Rönsch, Ulrich Betz, Jürgen Konradi, Janine Huthwelker, Claudia Wolf, Ruben Westphal, Philipp Drees

**Affiliations:** 1Department of Orthopedics and Trauma Surgery, University Medical Center of the Johannes Gutenberg University Mainz, Langenbeckstrasse 1, 55131 Mainz, Germany; johanna.kniepert@unimedizin-mainz.de (J.K.); philipp.drees@unimedizin-mainz.de (P.D.); 2Institute of Physical Therapy, Prevention and Rehabilitation, University Medical Center of the Johannes Gutenberg University Mainz, Langenbeckstrasse 1, 55131 Mainz, Germany; henriette.roensch@unimedizin-mainz.de (H.R.); juergen.konradi@unimedizin-mainz.de (J.K.); janine.huthwelker@unimedizin-mainz.de (J.H.); claudia.wolf@unimedizin-mainz.de (C.W.); 3Institute of Medical Biostatistics, Epidemiology and Informatics, University Medical Center of the Johannes Gutenberg University Mainz, Obere Zahlbacher Straße 69, 55131 Mainz, Germany

**Keywords:** back pain, spinal motion, rotational movement, dynamic surface topography, videorasterstereography

## Abstract

Current routine diagnostic procedures for back pain mainly focus on static spinal analyses. Dynamic Surface Topography (DST) is an easy-to-use, radiation-free addition, allowing spine analyses under dynamic conditions. Until now, it is unclear if this method is applicable to back pain patients, and data reports are missing. Within a prospective observational study, 32 patients suffering from thoracic and lumbar back pain were examined while walking, randomized at four speeds (2, 3, 4, 5 km/h), using a DST measuring device (DIERS 4Dmotion^®^ Lab). The measurement results were compared with those of a healthy reference group. We calculated the intrasegmental rotation for every subject and summed up the spinal motion in a standardized gait cycle. The Mann–Whitney U Test was used to compare the painful and healthy reference groups at the four different speeds. In a subgroup analysis, the painful group was divided into two groups: one with less pain (≤3 points on the Visual Analogue Scale) and one with more pain (>3 points on the Visual Analogue Scale). The Kruskal–Wallis Test was used to compare these subgroups with the healthy reference group. Of the 32 included patients, not all could walk at the intended speeds (5 km/h: 28/32). At speeds of 2–4 km/h, our results point to greater total segmental rotation of back pain patients compared to the healthy reference group. At a speed of 3 km/h, we observed more movement in the patients with more pain. Overall, we monitored small differences on average between the groups but large standard deviations. We conclude that the DST measuring approach is eligible for back pain patients when they feel confident enough to walk on a treadmill. Initial results suggest that DST can be used to obtain interesting therapeutic information for an individual patient.

## 1. Introduction

Back pain is still a widespread health problem. Besides affecting the patients’ quality of life and movement behavior, its influence constitutes a significant burden on social security finance [1]. Back pain is very complex, and various factors contribute to the situation, so it cannot be treated according to a standardized protocol. One problem concerning the determination of the correct therapy for back pain is the fact that its source is various and often unknown, leaving us with a high percentage of patients suffering from unspecific pain without any structural correlation [2,3].

Even if a specific diagnosis can be made, it is well known that a wide variety of pain conditions can be linked with different functional findings [4,5]. 

Current routine diagnostic methods, like X-ray and MRI, are usually static analyses, able to show structural damages and degenerations but not dysfunctions of joints or tissues (e.g., a joint blockage). That is why the focus on static structural alterations has limited explanatory power for unspecific back pain [6].

Hence, an appropriate measuring system should be able to analyze the spinal motion under dynamic conditions in a standardized way.

For example, Lamoth et al. [7] investigated the movement pattern of unspecific back pain patients with a marker-based optic measurement system. They could show that back pain patients walked at lower preferred speeds and had a more in-phase movement pattern between the thorax and pelvis. Their measuring procedure just allowed global angular measurements of the upper body and pelvis but not on a vertebral plane [7]. A recent meta-analysis revealed that subjects with persistent low back pain walked slower and with shorter stride lengths. Furthermore, they also showed a more in-phase movement pattern of the thorax and the lumbar spine [8].

However, as most motion capture systems work on a marker-capturing basis [9], until now, during gait analyses, spinal motion was examined mostly only by observing the motion of the thoracic or lumbar back as a block. Almost no segmental motion was mapped, and no relationship to the corresponding gait phases was drawn [7,10].

Dynamic surface topography (DST) might offer a possibility to derive segmental spinal motion from the surface of the back during gait. DST is an easy-to-use, radiation- and contact-free measurement opportunity, allowing further spine analyses not only under static but also under dynamic conditions. A pattern of horizontal, parallel lights is projected onto the subject’s bare back and distorted by the back’s curvature. That distortion is analyzed by triangulation and mathematical shape analysis due to a fixed angle between the camera and the light projector.

Using a mathematical algorithm, conclusions can be drawn from the surface topographic curvature picture to the underlying spine, providing a virtually constructed 3D position of each thoracic and lumbar vertebral body (from C7 to L4) and the pelvis during gait. The spinous process of L5 cannot be reflected adequately onto the skin; therefore, the section L4 to the pelvis is considered as one moving unit. A spine model, which was developed by Turner Smith in the 1980s, is used to create this 3D rendering [11,12,13,14,15,16,17]. Through comparison with X-ray or computed tomography (CT) images, the reliability [18] and validity [16,19] of static ST could be shown. A gold standard for detailed segmental motion analysis (e.g., dynamic X-ray-based measurements) is missing, so there exists no proven validity of DST. Nevertheless, based on the reliability, reproducibility [11,20], and accuracy [11,21,22] of DST, inter- and intraindividual comparisons can be made. For more detailed background information on the operation principle of DST, the reader is referred to [23,24]. Currently, there is no validation for DST in walking [25].

There are first results available, describing DST-captured spinal motion during gait in healthy reference cohorts [25]. For patients with back pain, such data are still lacking. In addition, it is not yet clear whether this method is at all suitable for patients with back pain, as it requires participants to walk on a treadmill without using the handrails. For this reason, the aim of this paper is to present our experience from a first study performing DST measurements with back pain patients and comparing the results for segmental spinal movement of the painful area to the healthy reference group.

## 2. Methods

### 2.1. Inclusion and Exclusion Criteria of the Study Population

Subjects suffering from acute or chronic back pain (specific or non-specific) with any pain intensity (at least 1 on Visual Analogue Scale (VAS, 0–10) in rest or motion) in the thoracic or lumbar spine region were enrolled. We did not differentiate between specific and non-specific back pain, as the focus was on the applicability of DST rather than the explicit study of back pain patients. The in- and outpatient recruitment was conducted at the University Medical Center of Mainz from September 2017 till January 2018. Furthermore, patients were informed about the study via flyers and word-of-mouth recommendations. Subjects were excluded if any of the following aspects applied to them: acute fractures; balance disorders or major gait abnormalities, which prevent freehand walking on the treadmill; and illnesses, which influence gait or balance (e.g., Mb. Parkinson, Multiple sclerosis, hemiparesis or -plegia, polyneuropathy).

To ensure patient safety walking on a treadmill with back pain, subjects had to perform the Two-Minute Walk Test (2MWT) [26] beforehand. The 2MWT was used to discover any major gait abnormalities (exclusion criterion), to ensure a walking speed of at least 2 km/h, and to determine the subject’s possible maximum speed on the treadmill.

Subjects were later excluded when they were unable to walk at least 2 km/h or freehand on the treadmill or due to surface alterations of the back (e.g., tattoos, big scars). The latter would lead to measuring errors of the DST.

Because of the contradictory results of other studies on the influence of BMI on the measurement results of DST [19,27], no participant was excluded concerning BMI. As the focus of the paper was on clinical applicability of DST and only in the second step on the collection of data from back pain patients in general, no additional imaging to distinguish the cause of back pain (specific or non-specific) was required to participate.

### 2.2. Measuring Device

For the examination of spinal motion during gait, the modified DIERS 4Dmotion^®^ Lab was utilized, consisting of the DIERS Formetric III 4D and the DIERS pedogait, using its own software DICAM v3.7 (Diers International GmbH, Schlangenbad, Germany).

The device measures more than 100 different parameters of the spine (C7-L4) and the pelvis motion with a frequency of 60 Hz. Rotation in the transverse plane was selected from this wide spectrum for the analysis in this study, as the reference values of the healthy comparison group are also available for this parameter [25,28]. In cooperation with the manufacturer, a device interface was built, with which all measured data are exported chronologically and further evaluated. The measured values of the trunk’s surface rotation at the height of each vertebral body (except for the fifth lumbar (L5)) could be mapped continuously onto the thoracic and lumbar spine. Timely synchronized data of an embedded foot pressure measuring plate (100 Hz) allowed the relation of the spinal model data to the gait cycle phases.

### 2.3. Measuring Procedure

Before the measurements were taken, the subjects completed a pain questionnaire and performed the 2MWT. The pain questionnaire included the location of the pain (thoracic and/or lumbar), the pain intensity (VAS (10 cm, 0–10)), and the BMI.

For the dynamic measurement, it was necessary to apply red light reflecting markers onto the anatomical landmarks of the vertebra prominens (VP) and the two dimples of Venus. Therefore, we used the static DIERS Formetric 4D average measurement to confirm or correct the marker’s accurate position. This procedure was also recommended by Betsch et al. [11] prior to obtaining dynamic measurements. To capture habitual gait, at each measurement, the participants had to walk on the treadmill for two minutes before the recording was started without an additional announcement. More information about the measuring setup and procedure can be found in the dissertation by JK [29].

With the Formetric III 4D, the subject’s bare back was recorded for eight seconds (at walking speeds of 2 and 3 km/h) and six seconds (at speeds of 4 and 5 km/h) to capture at least three gait cycles per measurement.

### 2.4. Healthy Reference Group

A group of 134 healthy subjects (average age: 39.81 years, standard deviation (SD): 12.45 years, body mass index (BMI) ≤ 30.0 kg/m^2^) was used for comparison [25,26]. The study on the reference data of the healthy subjects was carried out using the same measurement methodology as this study on back pain patients. Due to the interindividual variations in motion sequences described in other studies, healthy subjects were not matched in terms of age and sex in this study [30,31]. To be considered healthy, the subjects had to meet several prerequisites (no history of surgery or fracture between C7 and pelvis, no medical or therapeutic treatments between C7 and pelvis within the last 12 months and no due to musculoskeletal problems, in general, within the last 6 months) and pass several tests to demonstrate adequate gait stability (timed up-and-go test, 2MWT, back performance scale) ((WHO (INT: DRKS00010834)) [25,26].

### 2.5. Statistical Analysis

The number of subjects was determined in advance using a power analysis: With 29 patients, a difference of 1 standard deviation with a power of 90% to the 0.83% level can be demonstrated. In order to test the clinical applicability, the primary endpoint of the study was defined as how many subjects the measurements could be carried out on and, secondarily, at which walking speeds.

The raw data of the measurements were exported in relation to the phases of the gait cycles as separate files with DICAM v3.7 software and merged using SAS software (Version 9.4). The data of each measurement were set in relation to the duration of its gait cycles (0–100%). Afterwards, the three gait cycles per measurement were combined as one gait cycle as a smooth curve using a spline function. A total of 101 values were obtained using integer percentages (0–100%) found in the graph. This procedure of generating a standardized gait cycle per measurement allowed all the measurements of different lengths to be compared [32,33,34,35,36].

Starting at the VP, we calculated the segmental rotational motion for all the segments down to the pelvis with IBM SPSS Statistics (Version 23). The rotational position of a spinal segment was computed by subtracting the position value in degrees of one vertebral body in the transversal plane from the position of the vertebral body below at the same point in time. By subtracting the segment’s position between two consecutive time periods, the segment’s rotational motion was obtained. For each measurement and segment, the absolute values of the rotational motion per standardized gait cycle were summed up (sum of motion per gait cycle (SoMpGC)). The calculation concept for the SoMpGC is displayed in Figure 1.

Subjects could mostly only name a pain area and not a single painful spinal segment, which is why a distinction was only made between thoracic and/or lumbar pain. Depending on the location of the subject’s pain, the SoMpGC of the affected segments (thoracic and/or lumbar) was used to calculate the average total intrasegmental rotation for each spinal segment separately.

We collected data on pain characteristics (e.g., duration of pain, drug medication, and paresthesia) and anthropometric data such as height, weight, and age, which will be used in future analyses.

Despite the metric character of intrasegmental rotation and the sum of motion per gait cycle, we used non-parametric tests for group comparison because of the small sample size and a conservative approach to interpretation. We chose the Mann–Whitney U Test for comparison of the groups at different speeds. For subgroup analysis, the back pain group was divided by VAS values (median VAS 3.25) into a low-grade (VAS ≤ 3) and a high-grade (VAS > 3) pain group. The Kruskal–Wallis Test was used for group comparison. We used calculation of Cohen’s d with the respective pooled standard deviation for effect sizes [37,38]. Positive values point toward the painful areas, and negative values points toward the healthy reference group. All calculations were conducted with IBM SPSS Statistics (Version 29).

## 3. Results

### 3.1. Characteristics of the Study Population

Thirty-four subjects suffering from back pain (specific and non-specific) between C7 and L5 were recruited, of which the measurements of 32 subjects (18 women, 14 men) between the ages of 19 and 68 were evaluated. The process of enrollment and dropout is displayed in Figure 2, and the participants’ characteristics are displayed in Table 1. During the 2MWT, the subjects showed an average walking distance of 190.72 m (SD: 35.60). Compared to the corresponding age cohort (18–54 years) of Bohannon, Wang, and Gershon [27] for obtaining standard data (women, 183.0 m; men, 200.9 m), the participants with back pain did not show any difference in maximum possible speed. All of them had pain in rest and over 70 % reported back pain also while walking. On average, the subjects showed a pain intensity of 4/10 (see Table 2).

### 3.2. Applicability of Dynamic Surface Topography

The recruitment was made more difficult because not all patients with acute pain felt confident enough for the pre-examinations and later for walking on a treadmill without holding. When subjects were enrolled, only 1 of 34 enrolled subjects could not be examined with the device due to difficulty when walking on a treadmill while suffering from a high level of pain and a delay caused by technical problems of the device.

Seven single measurements at different speeds (5%) could not be analyzed by the software due to technical/software problems. For one subject, only one measurement at 2 km/h was possible. Due to technical problems, this measurement and the subject both dropped out. In total, 115 measurements of 32 subjects at speeds from 2 km/h to 5 km/h were evaluated.

The faster the treadmill speed, the more difficult it was for subjects with severe back pain to walk on it. So, 28/32 (85%) of the included subjects could be examined at a treadmill speed of 5 km/h. The evaluation of the subjects’ number in relation to the treadmill’s walking speeds is displayed in Table 3.

### 3.3. Data of Dynamic Spinal Rotation

#### 3.3.1. Analysis of the Entire Group

The subjects’ average segmental rotation in the transversal plane of the segments in the painful area is displayed as the average SoMpGC and is set in comparison to the average SoMpGC of the healthy reference group. For the patient group, only the data of the segments in the respective painful regions were included in the analysis (thoracic spine or lumbar spine/thoracic and lumbar spine).

Figure 3a–d show the SoMpGC of the back pain patients and the healthy reference group at all four speeds measured. At 2, 3, and 4 km/h, an increased total rotation of segments in the painful area is seen in relation to the same segments of the healthy reference group. In the upper thorax, this difference can be up to 2°, and in the remaining spine, the difference is less than 1°. At 5 km/h, the difference reverses in the lower spine (between T9 to Pelvis), with the total segmental rotation of subjects with back pain minimally less than the total segmental rotation of the healthy reference group. In the Mann–Whitney U Test, we found *p* ≤ 0.05 only for thoracic segments.

Effect sizes for 2 km/h were large in the painful upper thoracic segments (d = 0.9 for T3/T4 and T4/T5), whereas the smallest value was detected for the painful segment L4/pelvis (d = 0.3). These effect sizes decrease at speeds of 3 and 4 km/h in the upper thorax. In the segments T6/T7, the value turns negative to indicate a greater movement in the healthy segments. At a speed of 5 km/h, there are small effect sizes in all segments, changing from positive to negative values and reverting. For further information, see Appendix A.

Because of the very small groups for the painful areas, the location of pain was not taken into account in the calculation. Hence, due to this explorative approach, we regard the results with *p* ≤ 0.05 as a trend.

#### 3.3.2. Analysis According to Pain Intensity

The subjects with lumbar back pain were also examined based on their pain intensity. At the speed of 3 km/h, 27 subjects (of 30 subjects with lumbar pain, 3 measurements could not be analyzed) with a median pain of 3.25 (derived from VAS) were evaluated (see Figure 4).

The low-grade pain group revealed an inhomogeneous pattern of the SoMpGC. While in the upper thoracic spine, their SoMpGC values were constantly higher than those of the healthy controls, starting at the T6/T7 segment, a strict movement pattern can no longer be recognized. Rather, the extent of movement fluctuates between more and less movement compared to the healthy comparison group.

The high-grade pain group has higher SoMpGC values compared to the healthy reference group, except the segment T6/7. The Kruskal–Wallis Test found differences between the high-grade pain and the healthy cohort in the segments VP/T1 to T5/T6 (*p* < 0.05). Because of the small subgroups, we consider the results as a trend.

In the comparison of both pain groups, the SoMpGC values of the segments T5/6 and T6/7 are nearly the same. In all other segments, the movement values of the high-grade pain group are higher than those of the low-grade pain group.

## 4. Discussion

The study could show that Dynamic Surface Topography can be used to examine back pain patients when they feel confident enough to walk freehand on a treadmill. However, this only affects a part of the overall group. Patients with severe pain, in particular, were often reluctant to undergo the examination. Unfortunately, it is not known how large this group is and what its exact composition is. It could be interesting to see why these patients refused to participate and if they can be convinced under better conditions. In this case, further investigations have to ensure the patients’ safety, e.g., very slow walking on the treadmill while holding on, followed by a very cautious increase in demand. It is conceivable that, thereby, unconfident and anxious patients could be examined as well.

A possibility to further increase the number of suitable patients could be the use of additional safety devices such as safety straps. However, this option is limited due to the necessary view of the back during the DST measurement. If these approaches fail, the DST measurement does not appear to be a suitable method for objective and standardized functional testing. For this group, a different measurement method, for example in standing, must be found. But, if patients feel confident enough to walk on a treadmill despite having back pain, a DST measurement can be carried out.

### 4.1. Spinal Motion of Thoracic and Lumbar Pain Patients

Most of the literature describes an average total rotation of only 1–2° [39,40], with some studies reporting 4° to a maximum of 6° [41] in the lumbar spine. Unlike passive examinations, active examinations, such as walking, do not exhaust the maximum rotational range of spinal motion [42,43]. Feipel et al. reported more rotational movement with increasing walking speed, but this did not exceed 40% of the maximum range of motion [42]. Segment height and degeneration level also seem to be relevant for segmental motion [39,44]. Since DST measuring offers continuous observation of the rotational behavior, we decided to present the movement as SoMpGC. The limitation to the maximum range of motion would significantly reduce the value of the measurement for the functional evaluation. However, this means that our results cannot be directly compared with the results above [39,40,42]. In a study using the VICON system for investigating back pain patients during level walking at individual speeds, significant differences in the amplitude of rotation were found. Patients with low back pain used 25 to 50% less rotational ROM in the lumbar spine, depending on the corresponding reference values [45]. On the other hand, Crosbie et al. [46] reported little or no effect of back pain on the segmental motion of the lumbar spine during overground walking. In contrast, our results indicate, at least at speeds up to 4 km/h, a trend for more movement in the back pain patients compared to the healthy reference group. At 5 km/h, this effect is no longer detectable [46]. An important factor could be the walking speed. Subjects with low back pain prefer a slower walking speed, lower step length, and cadence [8], but the surprising effect was particularly noticeable at slow walking speeds. It is possible that our study shows hypermobile syndromes triggered by degeneration, for example. However, we observed intensified rotation, particularly in the more painful patients. In these patients, higher muscle tone can be expected as a result of the pain to guard the spine during walking [8,47,48]. This effect is more evident, especially at speeds higher than 4.6 km/h [7]. The study by Lamoth et al. reports even synchronous, in-phase movements between the pelvis and thorax in subjects with back pain, which may lead to less movement in the spine [7].

We observed an increased difference in rotational movement compared to the healthy reference group at lower walking speeds. Walking at lower speeds is uneconomic and more unstable than faster walking or preferred walking speed, which needs more neuromuscular activation [49,50]. In back patients, the activation of local and global muscles is altered [51], which leads to modified movement patterns [51,52]. This could be an explanation for our detected greater difference in rotational movement between the back pain and healthy participants. 

### 4.2. Limitations of the Measuring System and the Study Design

Since the DST measurement does not directly measure segmental mobility but derives the movement from the back surface, it is conceivable that the movement does not actually change, but only the back surface, e.g., an increased muscle tone on the surface is incorrectly interpreted as the movement of the spine. Examinations of back pain patients with electromyography during gait detected higher muscle activity in the swing phase and additionally positively correlated with higher pain intensity [53]. In principle, this effect of the influence of the muscle tone is also conceivable with the other measurement methods for movement analysis. However, the DST measurement could react particularly sensitively to changes in muscle tone, as the entire width of the back contour is used for the analysis, not just the line of dorsal processes. This argument is even more validate, that as mentioned above, back pain patients activate more global than local stabilizing muscles [51]. Possibly, the movement of subcutaneous fat could also be misinterpreted by the software as segmental movement, especially in patients with a BMI >29 [54]. The combination of systematic and random mistakes makes it difficult to integrate the exact relationship between soft tissue and bony motion in data calculations [55]. The accuracy of DST measurements with respect to rotation has been determined with a mean deviation of up to 3° [16,18] compared to X-ray images and a deviation of 1–1.5% compared to the Vicon system [11]. Regarding these results and the small amount of rotational motion of the individual vertebral bodies in relation to the error indicated in the literature, the difficulty of determining the exact individual vertebral body rotation by means of DST must be considered. To our knowledge, there is no validation of the DST in walking. Measuring a cohort of pain patients with different measurement systems with comparable data processing would be very interesting in this context. In addition, the repeated measurement of patients in different stages of pain would be very informative in order to better understand the impact of back pain on the individual spine motion, in the case of this study, on the segmental rotation or SoMpGC.

### 4.3. Implications of the Findings for Clinical Practice

Overall, in our analysis, the mean difference in SoMpGC between the back pain patients and the healthy subjects is small, especially in relation to the standard deviation in both groups. This is to be expected for the group as a whole, as the variability is already high in the healthy group, and the group of back pain patients in our study was small and not uniform. Hence, we regard our results as an indication of maybe altered movement patterns on the vertebral plane. Nonetheless, based on the proven reliability, reproducibility [11,20], and accuracy [11,21,22] of DST measurements, useful information can be obtained from the surface of the back for individual cases as to whether the patient is currently using their spine more or less intensively. This can be a valuable hint for the therapeutic process of Clinical Reasoning. It objectively represents the movement information that is currently usually collected subjectively in the visual examination, also of the back surface. The information can become much more valuable if numerous parameters are included in the analysis at the same time in the future, resulting in a movement pattern. However, this is a challenge for the classically established analytical methods. Artificial intelligence offers an opportunity here, which has already been used in initial work in connection with DST measurement [20,21,34].

## 5. Conclusions

In our study, we were able to successfully apply DST measurement to back pain patients. The study population was limited by acute pain, fear, and the necessity of walking on a treadmill for the examination. Because of this, the application is limited to a subgroup of the overall back pain collective.

Until now, the influence of muscle activity and soft tissue displacement on the measurement results is unclear. On average, the differences in the measurement results between back pain patients and healthy individuals are small, but the standard deviation is large. Nonetheless, information can be obtained on the movement behavior of individual patients, therefore, could be useful in follow-up visits.

This study showed there is still a significant need for research in segmental motion analysis of the spine; yet, given the tremendous importance of back pain in society, it should be pursued further. For future investigations, precautions are conceivable to increase the subgroup (e.g., slowly increasing walking speeds, the decision of the patient when the speed increases, and to which speed limits). The integration of more parameters into the analysis, could greatly increase the value of the analysis.

## Figures and Tables

**Figure 1 bioengineering-12-00289-f001:**
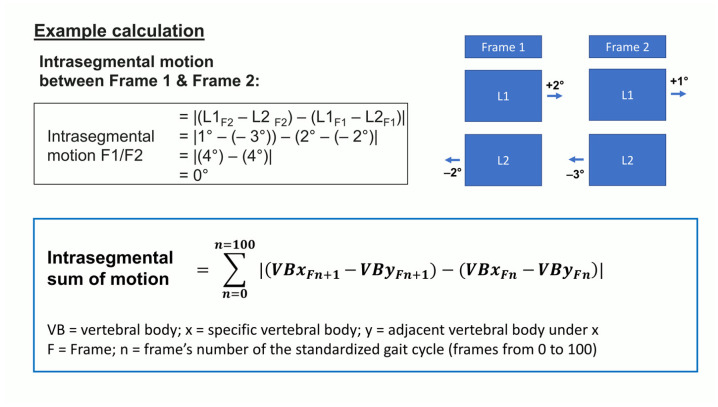
Example calculation for calculating intrasegmental rotation and sum of motion per gait cycle (SoMpGC).

**Figure 2 bioengineering-12-00289-f002:**
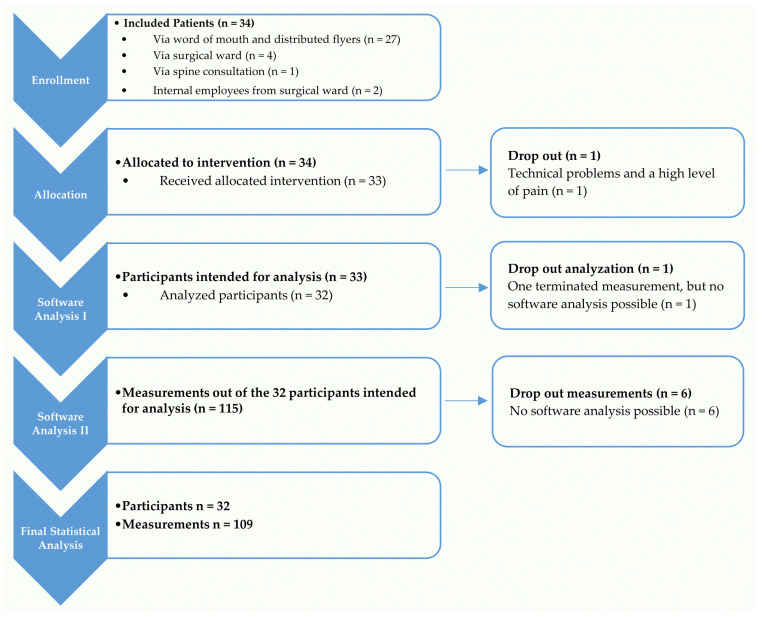
The process of enrollment, allocation, analysis, and reasons for dropout are displayed for the collective of subjects suffering from thoracic and/or lumbar back pain.

**Figure 3 bioengineering-12-00289-f003:**
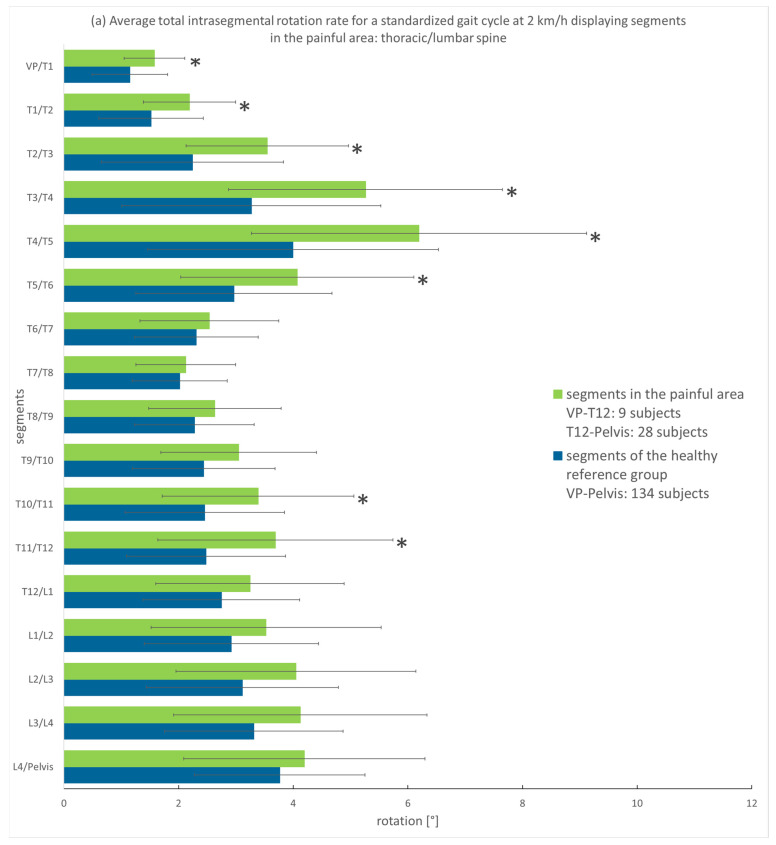
(**a**–**d**): Displayed are the mean and standard deviation of the SoMpGC in degrees (x-axis), displaying spinal segments (y-axis) located in the painful area thoracic or lumbar region (green) in comparison to the segments of a healthy reference group (blue): (**a**) at 2 km/h, (**b**) at 3 km/h, (**c**) at 4 km/h, (**d**) at 5 km/h; number of subjects in the painful area exceeds the maximum cohort number because subjects could have thoracic and lumbar pain. * *p* ≤ 0.05 (because of the small subgroups, we consider the results as a trend).

**Figure 4 bioengineering-12-00289-f004:**
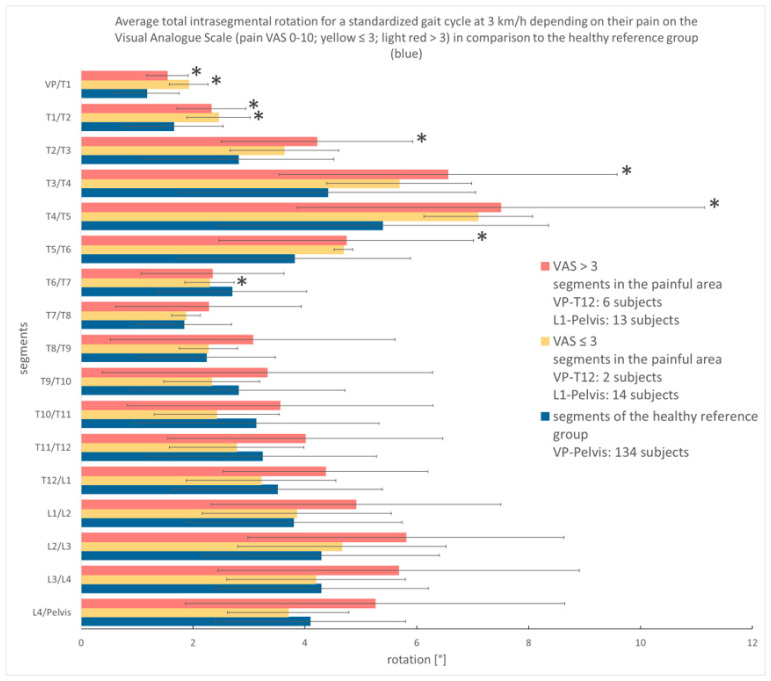
Displayed are the mean and standard deviation of the SoMpGC at 3 km/h in degrees (x-axis), displaying lumbar segments (y-axis) located in the painful area of low back pain depending on their pain on the VAS (Visual Analogue Scale) (pain VAS 0-10; yellow ≤ 3; red > 3) in comparison to the healthy reference group (blue); number of subjects in the painful area exceeds the maximum cohort number because subjects could have thoracic and lumbar pain. * *p* ≤ 0.05, comparison between the respective pain group and healthy reference group (because of the small subgroups, we consider the results as a trend).

**Table 1 bioengineering-12-00289-t001:** Characteristics of the analyzed participants: 1a. Display of the demographic characteristics; 1b. Display of the number and percentage of subjects and location of pain (thoracic and/or lumbar).

	Characteristics	Analyzed Participants (n = 32)
**1a**	Age (years): average (SD), range	44.53 (14.84), 19–68
	Male sex n (%)	14 (44%)
	BMI average (SD), range	26.01 (4.79), 16.76–37.56
	2MWT (distance in meters): average (SD), range	190.72 (35.60), 81.28–243.84
	2MWT (speed in km/h): average (SD), range	5.72 (1.07), 2.4–7.3
**1b**	**Location of back pain**	**Number of subjects (percentage)**
	Thoracic spine	2 (6%)
	Lumbar spine	23 (72%)
	Thoracic and lumbar spine	7 (22%)

Abbreviations: SD, standard deviation; BMI, body mass index; SD, standard deviation; 2MWT, two-minute walk test.

**Table 2 bioengineering-12-00289-t002:** Evaluation of the pain questionnaire: percentage of the subjects with rest pain and pain when walking, and average pain in terms of the average number of the subjects’ pain.

Pain Characteristics	Number of Subjects (Percentage)
Rest pain	32/32 (100%)
Pain when walking	23/32 (72%)
Pain level (VAS: 0–10)	average: 3.67/10 (minimum: 1, maximum: 8; SD: 1.83) median: 3.25; range: 1–8 SD: 1.83
**Time period of pain**	
<1 month	5 (15%)
1–6 month	4 (12%)
6–12 months	3 (9%)
12–24 months	6 (18%)
24–60 months	2 (6%)
>60 months	13 (39%)
**Pain medication**	
Yes	6 (18%)
None	27 (82%)
**Radiation to the legs**	
Right	5 (15%)
Left	9 (27%)
None	19 (58%)
**Paresthesia/Reduction in strength**	
Right	2 (6%)
Left	6 (18%)
Both sides	2 (6%)
Cervical spine	1 (3%)
None	22 (67%)

Abbreviations: VAS, Visual Analogue Scale; SD, standard deviation.

**Table 3 bioengineering-12-00289-t003:** Evaluation of the subjects’ number in relation to the treadmill’s walking speed.

Walking Speed on the Treadmill	Number of Subjects Examined (Percentage)
2 km/h	33 (100%)
3 km/h	31 (94%)
4 km/h	30 (91%)
5 km/h	28 (85%)

## Data Availability

The data can be made available upon request.

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
