# Peer review of "Dynamic Surface Topography for Thoracic and Lumbar Pain Patients—Applicability and First Results"

_bioengineering, 2025, doi:10.3390/bioengineering12030289_

Round 1
Reviewer 1 Report
Comments and Suggestions for Authors
the article is interesting and presents a very specific research question which, in my opinion, is worthy of scientific study, namely motion radiodiagnostics (in the case of the technique measured in the study: DST-captured spinal motion). I believe that the introduction does an excellent job of identifying the state of the art (the articles considered are recent) and explaining the research question. The research design, for this preliminary study, can be considered adequate (the authors, in the conclusions, correctly address the limits of the small size sample). The tables and figures are clear and explain the results well, the discussion of which is coherent in terms of content.
However, the article would benefit from a more formal and fluent English language. Furthermore, the format used does not allow a homogeneous distribution of the text, hindering readibility and overall presentation.
Comments on the Quality of English LanguageThe article would benefit from a more formal and fluent English language (e.g. "Till now" is a very informal way of stating "Until now"; "Its influence consititutes a significant burden on social security finance" instead of "it also has a severe economic impact on the social security systems"; "Given the complexity and ambiguity of this situation, we can state that the treatment of low back pain..." insted of "This complex and ambiguous situation means that the treatment of back pain is largely determined by...").
Reviewer 2 Report
Comments and Suggestions for Authors
Dear Authors,
I enclose the article with some minor revisions inside (Abstract, Methods, Results, and Conclusions) to improve its quality.

Reviewer 3 Report
Comments and Suggestions for Authors
There is no golden standard for the diagnosis of back pain. Dynamic Surface Topography DST offers an analysis method that can potentially improve the accuracy of diagnostics, particularly in combination with established methods. The ease of application and cost-effectiveness of the method and the technical realisation used here make it seem necessary to take a closer look at it. The application to a larger group of patients compared to a healthy reference group allows an assessment of the clinical possibilities of the method in patients with lumbar pack pain for the first time, even if there is no reliable information on the measurement uncertainty to date. The authors do not claim to want to build biomechanical models on their results, so that I consider the foreseeable limited quantitative reliability of the results obtained to be tolerable under the perspective of a desirable initiation of a discussion among experts. This discussion can initiate tests in clinical use, which can provide indications of further development needs and drive progress. The results presented offer a variety of approaches for a critical discussion.
Few criticism:
The use of non-parametric statistical tests in the group comparison in combination with the specification of the standard deviation in the bar charts is contradictory. An explicit statement should be made on this. As the aforementioned variables collected in parallel, such as BMI etc., which would explain the use of non-parametric tests, are no longer visible, an explanation should be provided on these and their use for the generation of the results presented - or otherwise only a brief reference should be made in the methods section to which possible confounder variables were collected. In the current state, there is a "gap" between methods and results. Minor detail: it would make sense to make the stars in the diagrams larger - only after drawing circles around these stars did the statements really jump out at me, with the immediately following consecutive formulation of contradictory hypotheses, especially regarding CLBP.
Reviewer 4 Report
Comments and Suggestions for Authors
Dear Editor,
Thank you for inviting me to review the manuscript "Dynamic Surface Topography for Back Pain Patients - Applicability and First Results" (ID: bioengineering-3481105). This study investigates the applicability of Dynamic Surface Topography (DST) for examining spinal motion in back pain patients and compares their movement patterns to healthy subjects. The main findings indicate that DST can be successfully applied to back pain patients who feel confident walking on a treadmill, with results showing increased total segmental rotation in pain patients at speeds of 2-4 km/h compared to healthy subjects, particularly in those reporting higher pain levels.
This manuscript addresses an important study topic in our understanding of objective spinal motion assessment in back pain patients. While the study provides valuable initial insights, several aspects require attention before publication.
General comments
- Theoretical framework. The introduction could better establish the gap in current diagnostic approaches and more clearly justify why DST might be particularly valuable for back pain assessment. The relationship between surface measurements and actual spinal motion needs more explanation.
- Methodology. The authors should provide more detailed justification for their measurement protocols and analytical approaches. The exclusion/inclusion criteria need clearer rationale, particularly regarding the decision not to control for specific vs. non-specific back pain. The validation of the DST measurement system for this specific application requires a more deepest discussion.
- Presentation and organization of results. Currently, the findings are somewhat fragmented between applicability and actual movement data. A more systematic organization would improve readability. The statistical analysis needs more robust justification, particularly regarding the small sample sizes in subgroups and the decision to treat results with p≤0.05 as trends rather than significant findings.
- Discussion section. While the authors acknowledge several limitations, the interpretation of findings could be more thoroughly integrated with existing literature. The clinical implications of the findings, particularly regarding the relationship between pain intensity and movement patterns, need more elaborate discussion.
Specific Comments
Introduction
- Expand the literature review on current diagnostic limitations in back pain assessment
- Provide stronger justification for using DST specifically for back pain patients
- Better explain the relationship between surface measurements and spinal motion
- Include more recent references on movement analysis in back pain patients.
Methods
- Clarify the rationale for not distinguishing between specific and non-specific back pain
- Provide more detail about the validation of DST measurements during walking
- Explain how potential confounding factors (e.g., muscle tension, subcutaneous fat) were addressed
- Include power calculations or justification for the sample size
Results
- Reorganize the results to more clearly separate applicability findings from movement data
- Provide a more detailed analysis of the relationship between pain intensity and movement patterns
- Include effect sizes for all comparisons
- Better justify the statistical approach, particularly regarding the treatment of p-values
Discussion
- Strengthen the interpretation of why pain patients showed increased movement at lower speeds
- Better explain the potential influence of muscle tension on surface measurements
- Expand on the clinical implications of the findings
- Include more comparison with other movement analysis methods in back pain assessment
Conclusions
- Make the conclusions more specific regarding the clinical utility of DST
- Better acknowledge the limitations of the current approach
- Provide more specific recommendations for future research
- Include clearer guidance for clinical implementation
In conclusion, the manuscript addresses an important topic and provides valuable initial insights into the use of DST for back pain assessment. However, substantial revision is needed to strengthen the methodological framework, clarify the results, and better establish the clinical relevance of the findings. Authors are encouraged to carefully consider and address all the points raised in this review to enhance their work's scientific quality and practical utility.
Kindly regards,
The reviewer
Reviewer 5 Report
Comments and Suggestions for Authors
Review of “Dynamic Surface Topography for back pain patients - Applicability and first results” by Johanna Kniepert, Henriette Rönsch, Ulrich Betz, Jürgen Konradi, Janine Huthwelker, Claudia Wolf, Ruben Westphal and Philipp Drees
This study explores the applicability of Dynamic Surface Topography (DST) in assessing spinal motion in back pain patients. The authors conducted a prospective observational study on 32 patients, comparing their segmental spinal rotation while walking at different speeds to a healthy reference group.
The article is poorly structured and presented, which gives the reviewer a negative impression of its content. Several key issues need to be addressed to improve clarity and rigor:
- Figure 1 lacks clarity and does not comprehensively illustrate the calculation concept. Figure 2 is poorly presented, reducing its effectiveness in conveying key information. The other figures present mean values but fail to highlight the large standard deviations, making it difficult to assess the significance of observed trends. It is strongly recommended to improving the figures with better annotations and higher resolution.
- The introduction is too short and does not clearly define its research question, making it uncertain what aspect of DST is being evaluated, its feasibility, diagnostic accuracy, or clinical decision-making potential. It is strongly recommended to provide a more structured introduction with a precise hypothesis and study objectives.
- The small and heterogeneous sample (mixing specific and non-specific back pain) introduces high variability, reducing the reliability and representativeness of the findings.
- While DST is validated for static spinal assessment, its accuracy in dynamic motion analysis remains uncertain.
- The study fails to address potential errors caused by soft tissue artifacts (muscle tone, fat movement), which may distort spinal motion measurements. A comparison with an established motion analysis system (e.g., VICON, fluoroscopy) would help validate the results.
- It is recommended to include a “Limitations” section.
The article is poorly structured and presented, which gives the reviewer a negative impression of its content. Numerous misprints and formatting issues further detract from its clarity and professionalism.
Round 2
Reviewer 4 Report
Comments and Suggestions for Authors
Dear Editor,
I have read the revised manuscript and the authors' detailed responses to my earlier comments. The authors have addressed most of my concerns in a detailed manner, and I applaud their diligent efforts in implementing the recommended changes.
The authors have greatly strengthened the manuscript by expanding the introduction's theoretical framework with more explanation for DST application, incorporating additional methodological details like sample size estimates, adding more effect size analysis to the results section, strengthening the discussion with better interpretation of the results, and reformulating the conclusion to be more clinically relevant.
The authors have effectively extended the introductory part with improved descriptions of the diagnostic limitations of back pain assessment. The additional discussion on static and dynamic analyses provides a more robust argument for using DST. Further, the incorporation of findings from some of the recent meta-analyses (Smith et al., 2022) reinforces the underlying literature.
The rationale for not differentiating between specific and non-specific back pain is now adequately explained. I am convinced by their argument, given the priority placed on applicability rather than diagnostic accuracy. The inclusion of power analysis and explanation of sample size adds much methodological strength. Their acknowledgment of the lack of validation of DST when walking as a limitation is appropriately noted.
The authors have enhanced their findings by a more detailed analysis of the association between movement patterns and pain intensity and by effect size estimation that conveys essential information regarding clinical importance. They have also provided a more detailed rationale for statistical method and p-value interpretation. I commend them for choosing to keep the current organizational framework, now with more descriptive subheadings, which separates findings of applicability from movement data.
The argument has been greatly enhanced with an improved explanation of why pain patients exhibited greater movement at lower speeds, an enhanced explanation of neuromuscular variables and changed movement patterns, the inclusion of a special section on clinical implications, and greater thoroughness in acknowledging limitations. The conclusions now more clearly express clinical utility, more appropriately acknowledge limitations, provide more specific recommendations for future research, and offer improved guidance for clinical application.
Although the manuscript is considerably improved, several minor aspects could be improved. The rationale for why surface measurements could reflect true spinal motion may be expanded a bit more. However, I appreciate the authors' argument that complete explanations can go beyond the nature of this paper. Although there is some discussion of possible confounding variables such as muscle tension and subcutaneous fat, coverage is quite limited.
In view of the extensive extent of changes and the meaningful enhancements provided in the manuscript, I suggest that this manuscript be accepted. The authors have made a diligent effort in responding to reviewer criticisms and have substantially enhanced both the scientific rigor and applied importance of their research.
Sincerely,
The Reviewer
Reviewer 5 Report
Comments and Suggestions for Authors
Since the authors have adequately addressed the concerns raised in the first round of review, I recommend accepting the paper.
PS: These are minor editorial issues that can be quickly resolved:
- On page 2, there are misplaced periods (.) at the end of sentences that need to be corrected.
- On page 2, a single quotation mark (" “ ") appears incorrectly.
- The placeholder text "Fig. 1 about here", "Fig. 2 about here" … should be removed or replaced with the correct in-text reference format used by the journal (e.g., "Figure 1 illustrates...").
Comments on the Quality of English LanguageThese are minor editorial issues that can be quickly resolved:
- On page 2, there are misplaced periods (.) at the end of sentences that need to be corrected.
- On page 2, a single quotation mark (" “ ") appears incorrectly.
- The placeholder text "Fig. 1 about here", "Fig. 2 about here" … should be removed or replaced with the correct in-text reference format used by the journal (e.g., "Figure 1 illustrates...").